# Bowel Ultrasound in Inflammatory Bowel Disease: How Far in the Grayscale?

**DOI:** 10.3390/life11070649

**Published:** 2021-07-04

**Authors:** Federica Furfaro, Arianna Dal Buono, Mariangela Allocca, Ferdinando D’Amico, Alessandra Zilli, Roberto Gabbiadini, Silvio Danese

**Affiliations:** 1IBD Center, Humanitas Research Hospital—IRCCS, 20089 Rozzano, Italy; arianna.dalbuono@humanitas.it (A.D.B.); mariangela.allocca@hunimed.eu (M.A.); ferdinando.damico@humanitas.it (F.D.); alessandra.zilli@humanitas.it (A.Z.); roberto.gabbiadini@humanitas.it (R.G.); silvio.danese@hunimed.eu (S.D.); 2Department of Biomedical Sciences, Humanitas University, 20090 Pieve Emanuele, Italy

**Keywords:** bowel ultrasound, inflammatory bowel disease, transperineal ultrasound, bowel thickness

## Abstract

Bowel ultrasound (BUS) is a non-invasive and accurate technique for assessing activity, extension of disease, and complications in inflammatory bowel diseases. The main advantages of BUS are its safety, reproducibility, and low costs. Ancillary technologies of BUS (i.e., color Doppler and wave elastography) could broaden the diagnostic power of BUS, allowing one to distinguish between inflammation and fibrosis. Considering the costs and invasiveness of colonoscopy and magnetic resonance, BUS appears as a fast, safe, and accurate technique. The objective measures of disease allow one to make clinical decisions, such as optimization, switch, or swap of therapy. Previous studies reported a sensitivity and a specificity of more than 90% compared to endoscopy and magnetic resonance. Lastly, transperineal ultrasound (TPUS) is a promising approach for the evaluation of perianal disease in Crohn’s disease (CD) and disease activity in patients with ulcerative proctitis or pouchitis. Bowel ultrasound is being incorporated in the algorithm of managing inflammatory bowel diseases. Transmural healing evaluated through ultrasonography is emerging as a complementary target for disease treatment. In this review, we aimed to summarize and discuss the current evidence on BUS in the management of inflammatory bowel diseases and to address the challenges of a full validation of this technique.

## 1. Introduction

Inflammatory bowel diseases (IBD) are chronic, progressive, and disabling conditions, characterized by a relapsing and remitting behavior and long-term complications (i.e., colo-rectal cancer and demolitive surgery) [1,2]. Dedicated physicians are becoming confident with the “treat-to-target” strategy in the management of IBD, aiming to prevent end-organ dysfunction [3,4]. For years, symptom control has been the primary therapeutic goal of IBD patients. However, clinical remission is not a reliable outcome for the optimal management of IBD. Indeed, one out of four patients who are clinically asymptomatic can have an endoscopically active disease. Conversely, even in the presence of endoscopic remission, symptoms continue to be reported [5,6]. In the need for objective and measurable endpoints, bowel ultrasound (BUS) has gained increasing relevance. Traditionally, ultrasound was not considered a valid method for the assessment of the small bowel and colon [7]. It has taken a long time since the first studies in the 1970s on the effectiveness of this technique for bowel examination and its recent scientific acknowledgement [7]. According to the Selecting Therapeutic Targets in Inflammatory Bowel Disease (STRIDE-II), transmural healing (TH), assessed by imaging techniques including bowel ultrasound (BUS), is considered a treatment target neither in Crohn’s disease (CD) nor in ulcerative colitis (UC) [4]. Nevertheless, especially in CD, transmural healing is an adjunctive outcome to endoscopic remission and might represent a state of deeper healing [4]. Current recommendations recognize BUS as a valid method for the assessment of the small bowel in newly diagnosed CD and, generally, for the monitoring of IBD [8]; however, a standardization of the intestinal and extraintestinal features of active disease is still needed. The main advantages of BUS are its non-invasiveness and low costs compared to computed tomography (CT) or magnetic resonance imaging (MRI) [9]. It has been recently demonstrated that when performed by a skilled operator, BUS has a comparable sensitivity and specificity to second-level techniques for assessing disease activity and complications of IBD [9,10,11]. Moreover, BUS is readily available and can be performed bedside by the dedicated gastroenterologist upon need (i.e., point-of-care BUS (POCBUS)) [12]. The monitoring through BUS of the bowel wall thickness (BWT) predicts the outcomes of IBD patients, particularly in CD for its transmural features [9,10,11]. In this review, we aim to examine and summarize the technical aspects and the current evidence on BUS in the management of IBD, focusing on the detection of disease activity, complications, and the newly emerging transperineal approach.

## 2. Technique and Features of Normality

Since the intestine is located superficially in the abdomen, the most detailed visualization of the bowel wall is acquired through a mid-frequency range transducer (5–10 MHz) micro-convex array [13], whereas the regular abdominal probes and the linear probes are low-frequency (1–6 MHz) and high-frequency transducers (10–18 MHz), respectively. The operator will assess the following main features of the intestinal tracts: wall thickness, wall border, echo pattern, vascularity, and motility. Several extraintestinal features belong additionally to the complete abdominal evaluation (i.e., lymph nodes, mesenteric fat, free abdominal fluid) [13,14].

According to the European Federation of Societies for Ultrasound in Medicine and Biology (EFSUMB) guidelines, the examination can be systematically performed with the aim of evaluating the whole intestine, starting from the hypogastrium, or left iliac fossa, firstly assessing the sigmoid colon, and then continuing along the colon to the terminal ileum, appendix, small bowel, and up to the stomach [13,14]. A fasting of 4–6 h is advisable, but not essential, in order to reduce the luminal content, the blood flow, and the peristaltic activity [13,14]. The iliopsoas muscle and the common iliac vessels can be used as landmarks to identify the sigmoid or the terminal ileum in the left or right iliac fossa, respectively. The normal bowel is stratified with five concentric layers that can be distinguished for their echogenicity (Figure 1): the most inner layer identifies the hyperechoic mucosa/lumen interface, while the most outer layer is an echogenic interface between the serosa and the confining organs or structures [15,16]. The BWT is the only fully quantitative ultrasound parameter that is measured from the external hyperechoic layer of the serosa to the internal hyperechoic interface between the lumen content and the mucosa [15,16]. To date, a bowel thickness of 2 mm was established by the EFSUMB guidelines as a threshold for the definition of normality [13]. In contrast, most studies and meta-analyses indicate a cut-off between 3 and 4 mm as a threshold of disease activity, especially for IBD patients [17,18]. In addition, a semi-quantitative grading of intestinal wall vascularity through the *Limberg score* has been described in the literature and is routinely used in clinical practice (Table 1) [18,19]. The evaluation of the rectum deserves to be discussed separately: the sensitivity of BUS in detecting a rectal location of IBD is approximately 15% [17,18]. Some recent evidence suggested a cut-off of 4 mm of BWT for the rectum, measured transperineally [20].

Regarding the evaluation of the bowel diameter, it can considerably vary (i.e., recent meals or fluid intake), but when the small bowel dilated segment becomes larger than 25 mm it is generally considered as abnormal, especially if a reduction in motility is observed, and a large bowel of more than 5 cm is also considered abnormal [14]. With respect to motility, the operator should assess any loss of elasticity and peristaltic movements [13,14]. Finally, among the evaluable extraintestinal features there are mesenteric fat, mesenteric lymph nodes, and abdominal free fluid [13,14]. An example of normal findings at bowel ultrasound of the sigmoid colon is shown in Figure 2.

## 3. Crohn’s Disease in Bowel Ultrasound

Crohn’s disease (CD) is a chronic, progressive disease that may affect any site of the gastro-intestinal tract, with a typical segmental/skip and transmural involvement [2]. The diagnosis and the monitoring of CD is based on the combination of clinical, laboratory, and endoscopic findings; histopathological reports; and imaging studies. There are known advantages in terms of detection of BUS over endoscopy in several cases, such as an incomplete colonoscopy, proximal locations of disease (i.e., distant from the ileo-cecal valve), and complications (i.e., fistulas, abscesses, and strictures) [9,10,11,12]. The indications to perform BUS in CD patients are summarized in Table 2.

The accuracy of BUS in CD assessment has been extensively demonstrated [17,19,21,22]. Several meta-analyses assessed the pooled sensitivity and specificity of BUS in CD by 88–89% and 93–97%, respectively [17,22], thus demonstrating that the detection of active disease in CD patients is precise and reliable, particularly for locations of disease in the small bowel [17,22]. Therefore, the ultrasonographic examination can be used to monitor disease activity and response to medical treatments [17,22].

### 3.1. Ultrasound Features of CD

A cut-off of bowel thickness greater than 3 mm is commonly adopted to predict disease activity with a sensitivity of 88–89% and a specificity of 93–96% [17,18,22]. Interestingly, a cut-off of 4 mm has a lower sensitivity (75%) despite a higher specificity (97–98%) [17]. When assessing the BWT of the colon, it is the most precisely determined when avoiding the haustrations [14,15,16]. Moreover, the longitudinal extent of disease has to be measured [14,15,16]. A lack of compressibility by the transducer and the loosening of the normal wall stratification can be observed in active CD [14,15,16]. In subjects with acute CD, the bowel wall appears hypoechoic, reflecting the corresponding oedema of the tissue infiltrate; in case of severely active diseases, it is possible to visualize the presence of deep mural ulcers that can additionally disrupt the stratification of the bowel wall [14,15,16]. With this regard, several studies proved that the loss of mural stratification is associated with clinical and biochemical activity, as well as with histological activity, and with an increased risk of surgery in CD [19,23,24].

As mentioned above, a semi-quantitative assessment of bowel wall vascularity using color Doppler imaging gives a complementary estimation of disease activity: the vascular patterns correlate with clinical and endoscopic activity [25,26]. In detail, the *Limberg score* (Table 1) is associated with the clinical activity, estimated through the Crohn’s disease activity index (CDAI), with a sensitivity of 82% (*p* = 0.01) [25]. The same study revealed a statistically significant association between the histological activity and the vascularity assessed at BUS (*p* = 0.03) [25]. Concerning endoscopy, considerable correlation (correlation coefficient r = 0.70, *p* < 0.001) was detected between the *Limberg score* and the simple endoscopic score for Crohn’s disease (SES-CD) at colonoscopy of 108 CD patients [26].

Furthermore, BUS allows the evaluation of several extraintestinal, indirect features of disease. In detail, enlarged loco-regional mesenteric lymph nodes are commonly encountered at BUS [15,16]. Inflammatory lymph nodes related to active CD have an oval shape and appear hypoechoic, with a diameter less than 5 mm and a short axis less than half of their longitudinal diameter [13,14]. This finding is less specific than BWT, echo pattern, and vascularity in terms of prediction of disease activity, and it could be linked to young age, early disease, and with the presence of abscesses or fistulae [27]. Mesenteric fat hypertrophy or creeping fat is an additional parameter of inflammation: the typical aspect is hyperechoic, almost “solid” [13,14,28]. Mesenteric fat hypertrophy has been less extensively investigated compared to all other sonographic findings, and it is known to be correlated with the clinical biochemical activity of CD [28]. Extraintestinal BUS findings are shown in Figure 3. Free fluid is a further common and reproducible BUS finding, generally found close to the inflamed bowel tract. It seems to be rather unspecific since it is commonly encountered in several non-IBD conditions [14].

Lastly, it has been extensively demonstrated in retrospective studies that BUS is able to detect and predict an early surgical recurrence after ileo-colonic resection [29,30,31].

### 3.2. Complications of Crohn’s Disease

In the natural history of CD, abdominal complications, such as stenosis, fistulae or abscesses, and, more rarely, free perforation, can occur [2]. In these cases, a prompt diagnosis is desirable since the management often involves surgery [8].

A recent prospective comparative study conducted on a cohort of CD patients demonstrated that BUS has a sensibility and a specificity in detecting strictures, fistulas, and abscesses located in the terminal ileum of 88–100% and 90–98%, respectively [32]. A lower sensitivity of this technique was observed with respect to colonic segments (76%) [32]. At US examination, a bowel stenosis is characterized by thickened walls with associated narrowed lumen (less than 10 mm) and dilatation of the proximal loop of 25–30 mm [14,15,16]. Additionally, hyperperistalsis of the pre-stenotic intestinal tract can be observed [14,15,16]. In general, in our clinical practice, the chronic stenosis has a particular disposition of the material inside the dilatated loop: it is solid on the bottom and fluid on the top. A recent systematic review confirmed BUS as a highly precise technique for the diagnosis of stenosis: the estimated sensitivity ranged from 80 to 100%, and the specificity varied from 63 to 75% [33]. With respect to stenosis, there is an open debate whether BUS is able to distinguish between a predominantly inflammatory and a fibrotic stricture. The relevance of this issue consists in a substantially different management: patients with evidence of a prominent inflammation might benefit from medical treatment, whereas patients with evidence of a fibrotic stricture would rather be advised for surgery (i.e., strictureplasty, resection) or endoscopic dilation [34]. It has been proven that a purely inflammatory stenosis would appear hypoechoic and highly vascularized, while a preserved stratified echo pattern can indicate fibrosis with no or poor signal at color Doppler [23,34]. Nevertheless, in real clinical practice, the stenosis is rather composed at the same time by an inflammatory and a fibrotic component: this explains the heterogeneity and inconclusiveness of the many studies on this topic, even when the stricture features are assessed through contrast-enhanced ultrasound (CEUS) or elastography [23,35,36].

A penetrating CD can be complicated by fistulas that are visualized as hypoechoic tracts originating from the intestinal wall either with a blind end or rather in continuity with mesenteric structures and confining organs (i.e., entero-mesenteric, entero-enteric, entero-vaginal, entero-vesical). The sensitivity of BUS in detecting a fistulizing disease is lower than for other complications of CD and has been assessed by 67–87% [37].

Conversely, the abscess is a purulent collection; the absence of gaseous material often allows one to distinguish fistulas from abscesses [13,14]. The typical sonographic appearance of an abdominal abscess is a hypo or an-echoic lesion containing gaseous (seen as bubbles) and liquid material, having often irregular margins and a posterior wall enhancement [13,14]. A further distinction must be made between an abscess and an inflammatory mass [38]. The latter is frequently highly vascularized and presents a diffusely increased enhancement at contrast-enhanced ultrasound (CEUS), while an abscess would enhance only in the periphery, with a typical avascular center [38]. BUS has been demonstrated to be similarly accurate as computed tomography (CT) and magnetic resonance (MRI) in diagnosing abdominal abscesses in patients with CD [39]. In more detail, according to a recent systematic review, the sensitivity and specificity with this latter specific indication ranged from 81 to 100% and 92 to 94%, respectively [39]. Notably, the detection of an abdominal abscess in patients treated with biologic agents demands a temporary discontinuation of the therapy and contraindicates an eventual therapy start, highlighting the relevance of the ultrasound monitoring [8].

### 3.3. Transperineal Ultrasound

Transperineal ultrasound (TPUS) allows one to evaluate the distal rectum, anal canal, and the perianal tissues that are not visualized in the transabdominal examination [38]. Indeed, the sensitivity of BUS in detecting disease activity of the rectum can be as low as 15% [40]. TPUS is easy and noninvasive compared to endo-rectal/anal approaches. The evidence concerning the accuracy of TPUS in IBD is still rare, and its clinical use is only emerging in very recent years. The exam is performed with the patient on the left lateral decubitus and with bent legs [40]. The main indications of TPUS are any known or suspected fistulas or collection in the perianal region in CD patients. In these patients, the use of intravenous contrast can improve the assessment of perianal abscesses, allowing a better differentiation from inflammatory masses and fistulas [41]. Several anatomical landmarks can be identified: the anal canal, internal and external anal sphincters, symphysis pubis, urinary bladder, prostate, and vagina [41].

In detail, the location of the fistula/collection should include the site of the anal canal (i.e., inner third, middle, or outer third), and the site on a clock representation is where 12 o’clock corresponds to the anterior wall of the anus [42]. Fistulae of the perianal region are classified according to Parks classification, which summarizes the anatomical course of the fistula in relation to the sphincters [43]. The Parks classification is reported in Figure 4. Among the available data, Mallouhi et al. firstly reported a sensitivity of 100% and a specificity of 94–100% of TPUS in detecting perianal fistulas and abscesses in a cohort of 62 IBD patients [44].

A recent systematic review with meta-analysis by Maconi et al. showed that TPUS has high sensitivity in detecting and classifying perianal fistulas (98.3 and 92.8%, respectively) [45]. A comparable accuracy was also found for the detection of perianal abscesses (sensitivity of 86.1%) [45]. Perianal abscesses can vary in size and shape and are classified as pelvi-rectal, inter-sphincteric, ischiorectal, and superficial perianal abscesses [45]. Despite the above-mentioned studies, pelvic MRI remains the preferred and recommended radiologic modality for the most detailed imaging of perianal CD [8]. Figure 5 shows examples of TPUS findings. Considering that a pelvic MRI costs on average USD 550 (range 500–1000), these high costs might be overcome by TPUS in the future.

## 4. Ulcerative Colitis

Ulcerative colitis (UC) is typically a mucosal disease rather than transmural; the gold standard for the diagnosis and monitoring is endoscopy [3,7]. Despite an initial disbelief due to the nature of inflammatory involvement in UC, BUS is emerging as accurate both in detecting active disease and assessing the extension of active UC [13,14]. Moreover, BUS addresses the disadvantages of endoscopy, such as its invasiveness and costs [13,14].

As reported by Smith et al. in their systematic review on the topic, BUS appears valuable in the routine assessment and management of patients with UC [46].

Common findings at BUS include the BWT, the loss of haustration and stratification, mesenteric changes, presence of lymphadenopathy, and an irregular mucosal surface due to post-inflammatory polyps and/or deep ulcerations [47].

Data from comparative and prospective studies have shown a strong correlation between wall thickness (>3 and 4 mm) and colonic vascular flow with C-reactive protein values and the endoscopic score [9,10,41]. These studies have led to the full validation of an ultrasonographic score, the Milan ultrasound criteria, that can be easily calculated (1.4 × bowel thickness (mm) + 2 × vascular flow) and indicates activity when ≥6.3 [10,11]. In contrast with CD, in UC patients, the loss of bowel wall stratification with a hypoechoic pattern is rarely observed; when present, it is associated with a severe disease [45].

Additionally, BUS may also be adopted for assessing response to treatment, in terms of reduction of the wall thickness [48]. A reduction in BWT of ≥2.5 mm has been proposed as sonographic response to therapy and is able to predict the clinical remission at a one-year follow-up [49].

Lastly, since the rectum is not always visualized with the transabdominal approach, TPUS has been proposed as a more accurate method with this respect. Indeed, TPUS has been investigated in UC with a good correlation between the rectal bowel wall thickness and *Limberg score* with the rectal Mayo score at endoscopy and histological scores (i.e., Nancy index) [20]. In this last study, a BWT ≥ 4 mm predicted endoscopic activity more accurately when evaluated through transperineal ultrasound compared to transabdominal ultrasound with a sensitivity and specificity of 100 and 45.8%, respectively (*p* = 0.0002) [20].

## 5. Discussion and Future Perspectives

This review elucidates the technical aspects and the current evidence on BUS in the management of IBD. The STRIDE-II recommendations, though recognizing BUS as a valuable method to assess the degree of inflammation in IBD, do not include TH as a treatment target either in CD or UC and is rather considered as an adjunctive target [4]. The meaning of “adjunctive target” is still under investigation and needs clarification. In our view, an adjunctive value of BUS in the management of IBD finds place, for example, in the decision of optimizing the therapy with biologic agents. Certainly, changing the therapy line exclusively on the basis of BUS findings might lead to a precocious withdrawal of the treatment, which is not advisable in view of the limited available therapy lines. Still, the possibility, offered by BUS, of frequent assessments allows a tempestive and anticipated optimization of the therapy without waiting for the endoscopic assessment, which could be wisely postponed.

To date, a growing body of evidence has been accumulating on the predictive value of TH, evaluated through BUS on the long-term outcomes [50,51,52]. Indeed, it has been demonstrated that sonographic remission evaluated after one year of anti-TNF therapy was associated with a longer remission without the need for a therapy change and a reduced need for surgery [51,52].

A further advantage of BUS, specifically compared to endoscopy, consists in assessing the entire gastrointestinal tract allowing, when suspected, the prompt recognition of complications (i.e., abscesses, stenoses, and fistulas) of CD and the subsequent prompt referral of the patient to surgeons. Notably, BUS is crucial in accelerating this process; however, it would substitute MRI in the pre-surgical evaluation.

Beyond these indications, the role of BUS has been increasingly broadened in recent years from the evaluation of disease activity and its complications toward the monitoring of disease progression and treatment response both in CD and UC [53,54]. Recent data endorse the adoption of BUS in the tight monitoring of IBD patients due to its accuracy in assessing signs of response and TH [53,54]. Particularly in the management of CD, BUS can be incorporated as a “bridge” examination to colonoscopy, since the reduction of the BWT accurately predicts the endoscopic response [53,54].

BUS is increasingly gaining relevance also in UC, especially considering the high costs of endoscopy. However, even though the role of BUS in UC also comprehends disease monitoring, it is to a lesser extent considering the unsubstituted value of colonoscopy in the diagnosis of cytomegalovirus (CMV) infection and in the surveillance for dysplasia/colon-rectal tumor.

In its route toward standardization, an evidence-based assessment through BUS has latterly been defined by an expert international panel of gastroenterologists and radiologists [21]: these efforts are going to endorse the use of BUS in future clinical trials as a substitute or alongside of endoscopy. In this survey, among the statements of greatest agreement for a good quality BUS assessment there were the cut-off of 3 mm for BWT both for the colon and small bowel, the use of the semi-quantitative *Limberg score* for vascularity, the need of acquiring an image of the rectum, and the description of the loss of stratification and/or the submucosal prominence [55].

An important matter of debate is the training of dedicated gastroenterologists: indeed, validated training times and acquirable skills are lacking, and there is no consensus on the definition of an “expert” BUS operator.

In the future, the combination of BUS and biomarkers, primarily fecal calprotectin, would largely substitute the more costly monitoring techniques (endoscopy and MRI) for their reliability in decision making.

Despite the gathered evidence, BUS has been widely underemployed in many countries outside of Europe. In particular, in North America (i.e., USA) this was due to a lack of local expertise and to less available training programs [56]. Moreover, there was historically a rooted skepticism on the clinical application of BUS as well as reimbursement matters [56].

As presented in the text, CEUS is an additional helpful tool in the characterization of suspected abscesses and inflammatory phlegmons, as well as in confirming the route of a fistula and quantitatively determining disease activity in IBD. Concerning the distinction between fibrotic and inflammatory strictures in IBD, there is encouraging evidence that CEUS integrated with further ultrasonographic tools would soon be determinant in this distinction.

Finally, the newly emerging TPUS represents a valid technique with possible future applications in the monitoring of operated IBD patients with ileal pouch, who more frequently receive medical treatment in the long term. Whether TPUS might be accurate in the monitoring of therapy response in pouchitis warrants dedicated prospective studies.

As far as we are concerned, despite the emergence of histology as a powerful target for better long-term outcomes, BUS would gain a well-defined place in the management algorithm of IBD due to its non-invasiveness, cost-effectiveness, readily availability, and reproducibility. The future challenges of this diagnostic modality consist of gaining increased accuracy for proximal disease (i.e., duodenum, jejunum, etc.) as well as for colonic segments.

In the upcoming era of more and more rigorous therapeutic targets, such as TH and histological healing, BUS can become the main instrument for a tailored monitoring and management strategy allowing one to anticipate and drive clinical decisions.

## Figures and Tables

**Figure 1 life-11-00649-f001:**
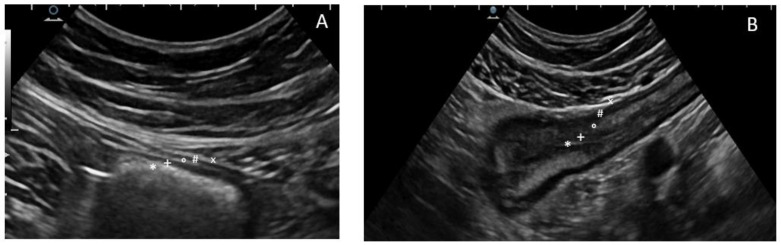
The five concentric layers of a normal (**A**) and thickened bowel (**B**) wall at ultrasonography. The most inner layer identifies the hyperechoic mucosa/lumen interface (*), then the hypoechoic mucosa (+), the hyperechoic sub-mucosa (°), and the hypoechoic muscularis propria (#), while the most outer layer is an echogenic interface between the serosa and the confining organs or structures (×).

**Figure 2 life-11-00649-f002:**
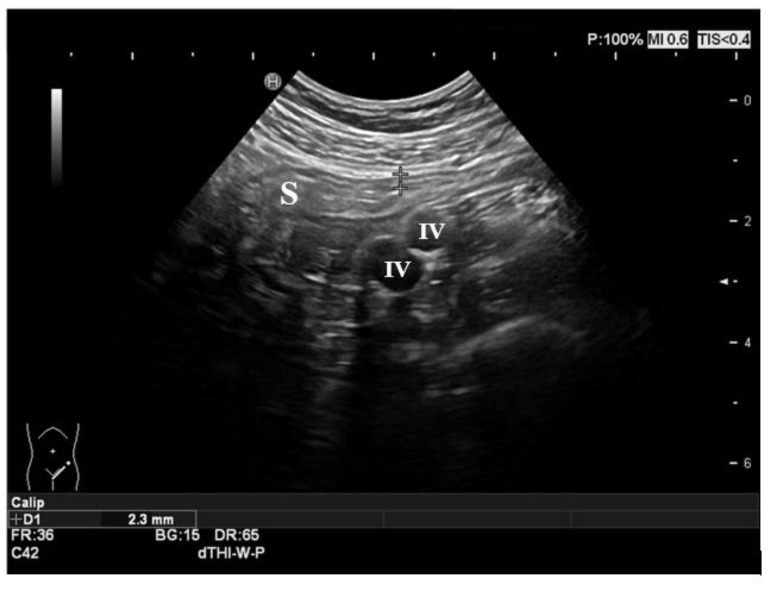
Normal ultrasonographic features of the gut. The sigmoid colon is shown in the figure, with normal thickness and stratification of the layers. The iliac vessels represent the anatomic landmark in the left iliac fossa. S: sigmoid colon; IV: iliac vessels.

**Figure 3 life-11-00649-f003:**
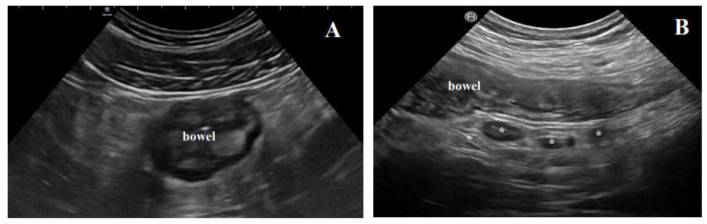
Extraintestinal findings of active disease at BUS. (**A**) The typical hyperechoic, almost “solid” appearance of mesenteric fat hypertrophy; (**B**) inflammatory lymph nodes (*) in a Crohn’s disease patient with active disease.

**Figure 4 life-11-00649-f004:**
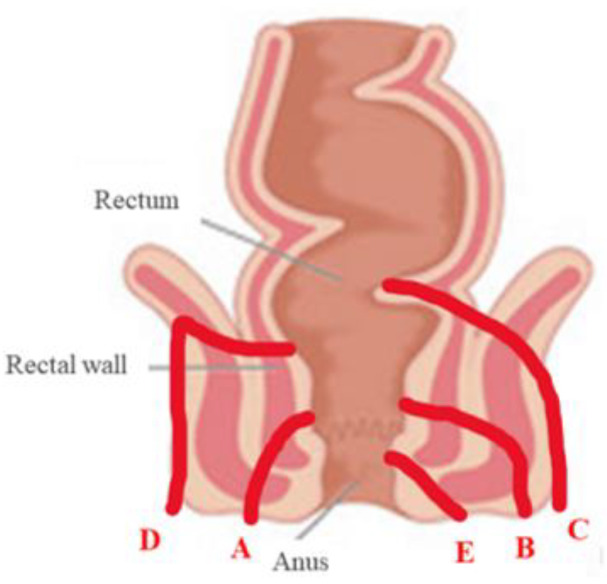
Parks classification of perianal fistulas (40). A: inter-sphincteric; B: trans-sphincteric; C: extra-sphincteric; D: supra-sphincteric; E: superficial perianal fistula.

**Figure 5 life-11-00649-f005:**
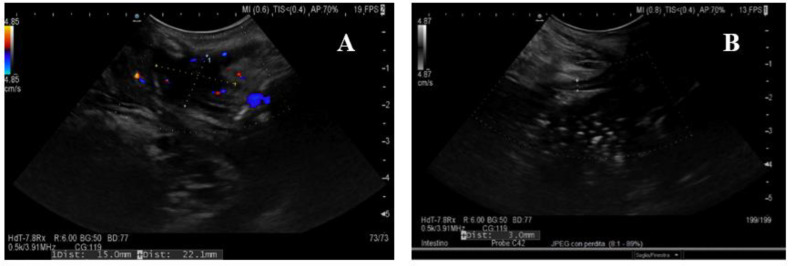
Ultrasonographic findings of transperineal ultrasound (TPUS). (**A**) A perianal abscess in an operated CD patient with typical peripheral color Doppler signal. (**B**) The wall of an ileal pouch is measured with the transperineal approach. CD: Crohn’s disease.

**Table 1 life-11-00649-t001:** Semi-quantitative assessment of vascularity through Limberg score (19).

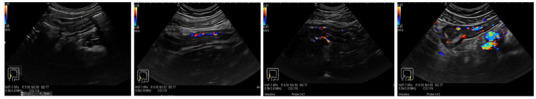
**Grade 1**	**Grade 2**	**Grade 3**	**Grade 4**
No vascularization signal at color Doppler	Mild: minimal signal, short stretches of vascularity in spots	Moderate: longer stretches of vascularity, blood vessels located only intra-mural	Severe: long continuousintra- and extra-mural blood vessels, extending into the mesentery

**Table 2 life-11-00649-t002:** Indications to perform BUS in Crohn’s disease.

Indications to Perform BUS in Crohn’s Disease
Initial work-up in suspected CD (i.e., differential diagnosis)
Baseline evaluation of disease activity and extension before therapy
Suspected complications (i.e., fistulas, abscesses, strictures)
Monitoring after/during the treatment course (response vs. worsening)

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
