# Peer review of "Bowel Ultrasound in Inflammatory Bowel Disease: How Far in the Grayscale?"

_life, 2021, doi:10.3390/life11070649_

Round 1

Reviewer 1 Report

Dear Authors,

I would like to thank you for the possibility to review your nice article. 

However, I do not see much novelty and scientific soundness. There are plenty of similar articles written in systematic review or meta-analysis manner. You should try rewriting your article as a systematic review

Author Response

Thank you for Your comment. We appreciate Your suggestion and agree that a systematic review would have a different scientific weight. However, we were asked to write a narrative review. Since this was the indication and request by the Editors of Life, we will leave the paper as narrative.

Reviewer 2 Report

In this review authors summarize and discuss the current evidence on bowel ultrasound in the management of inflammatory bowel diseases and to address the challenges of a full validation of this technique.

Author Response

Thank you for Your comment. We are glad to know that You appreciated our paper.

Reviewer 3 Report

This is a good and interesting critical review about the usefulness of noninvasive Bowel ultrasound techniques to study complications related to inflammatory bowel diseases. The review is well written, and in the text, the author evaluates the capacity of this technique compared to other techniques habitually used as endoscopy or magnetic resonance.  In the manuscript, the authors also discuss the future perspectives of the noninvasive Bowel ultrasound techniques. Consequently, I consider that the article could be published in life 

Author Response

Thanks for Your comment. We are glad to know that You appreciated our paper.

Reviewer 4 Report

  • Discuss the different use of BUS in different European regions and in USA
  • “a bowel thickness of 2 mm was established by the EFSUMB guidelines as threshold for the definition of normality”

What about the rectum?

  • Discuss the role of contrast in BUS
  • Discuss the prediction ability of surgical recurrence of BUS (see “Cammarota T et al. Role of bowel ultrasound as a predictor of surgical recurrence of Crohn's disease. Scand J Gastroenterol. 2013 May;48(5):552-5. doi: 10.3109/00365521.2013.777774. Epub 2013 Mar 11. PMID: 23477675.” “Power Doppler sonography to predict the risk of surgical recurrence of Crohn's disease. J Ultrasound. 2014 Jun 12;18(1):51-5. doi: 10.1007/s40477-014-0101-x. PMID: 25767640; PMCID: PMC4353824.” “A preserved stratified pattern of the bowel wall 1 year after major surgery does not influence the surgical recurrence of Crohn's disease. Ir J Med Sci. 2016 Feb;185(1):269-70. doi: 10.1007/s11845-015-1249-x. Epub 2015 Jan 13. PMID: 25579769.”)
  • In which clinical setting would you use BUS in UC?
  • “A further advantage of BUS, specifically compared to endoscopy, consists in assessing the entire gastrointestinal tract allowing, when suspected, the prompt recognition of complications (i.e., abscesses, stenoses and fistulas) of CD and the subsequent referral of the patient to surgeons.”

Without magnetic resonance enterography?

Author Response

Discuss the different use of BUS in different European regions and in USA.

Thanks for Your comment. As You suggested, we have added a dedicated paragraph on this in the discussion section.

“a bowel thickness of 2 mm was established by the EFSUMB guidelines as threshold for the definition of normality” What about the rectum?

Thank you for Your comment. We added few comments on the rectum, accordingly.

Discuss the role of contrast in BUS

Thank you for Your comment. CEUS is indeed mentioned over the text several times (see complications od CD, and trans-perineal US). As You suggested we discussed CEUS in more details in the discussion section.

Discuss the prediction ability of surgical recurrence of BUS (see “Cammarota T et al. Role of bowel ultrasound as a predictor of surgical recurrence of Crohn's disease. Scand J Gastroenterol. 2013 May;48(5):552-5. doi: 10.3109/00365521.2013.777774. Epub 2013 Mar 11. PMID: 23477675.” “Power Doppler sonography to predict the risk of surgical recurrence of Crohn's disease. J Ultrasound. 2014 Jun 12;18(1):51-5. doi: 10.1007/s40477-014-0101-x. PMID: 25767640; PMCID: PMC4353824.” “A preserved stratified pattern of the bowel wall 1 year after major surgery does not influence the surgical recurrence of Crohn's disease. Ir J Med Sci. 2016 Feb;185(1):269-70. doi: 10.1007/s11845-015-1249-x. Epub 2015 Jan 13. PMID: 25579769.”)

Thank you for Your comment. We agree that this is an important topic. As You suggested, we mentioned the data on surgical recurrence in BUS in the Crohn’s disease section. 

In which clinical setting would you use BUS in UC?

Thank you for Your comment. In the dedicated paragraph about UC we explained and discussed the role of BUS in detecting the disease activity, assessing the extension of disease and monitoring. In the discussion section we made the example of the importance of BUS in rapidly deciding to optimize the therapy.

“A further advantage of BUS, specifically compared to endoscopy, consists in assessing the entire gastrointestinal tract allowing, when suspected, the prompt recognition of complications (i.e., abscesses, stenoses and fistulas) of CD and the subsequent referral of the patient to surgeons.”Without magnetic resonance enterography?

Thank you for Your comment. We specified that BUS would accelerate the process of referring a patient to the surgeon without ever substituting MRI.

Round 2

Reviewer 1 Report

Dear authors, 

the manuscript has improved.

However, I do not feel it is suitable for the journal.